# PROGRESSIVE REVERSE UNDERSTANDING IMPROVES TEXT-TO-SQL GENERATION

## ABSTRACT

Recent advances in Text-to-SQL have significantly improved natural language interfaces to databases. Despite this progress, existing approaches still exhibit limitations of dependence on high-quality prompts and costly task-specific data, underscoring the need for more efficient and adaptable solutions. In this paper, we propose Progressive Reverse Understanding (**PRU**), a novel training paradigm that incrementally enhances Text-to-SQL generation by encouraging models to reason in reverse: from SQL back to natural language. PRU decomposes the learning process into progressive stages where models first learn to construct user intents from structured SQL, and then iteratively refine the mapping between language and structured queries. This bidirectional learning enables deeper semantic alignment while alleviating reliance on prompt quality and reducing the need for additional large-scale task-specific data by leveraging the data pairs inherent in the dataset. Extensive experimental results on the Spider dataset with open-source language models demonstrate the effectiveness of our approach in enhancing Text-to-SQL performance. Moreover, ablation studies and detailed analyses highlight the critical role of progressive reverse understanding in improving both syntactic correctness and semantic fidelity.

## 1 INTRODUCTION

Large language models (LLMs) demonstrate powerful generative capabilities and achieve remarkable performance across a range of linguistic tasks (Meta-AI, 2024; OpenAI, 2024; DeepSeek-AI, 2025). However, their capabilities in performing structure understanding and generation tasks still present substantial challenges (Cao et al., 2025; Lu et al., 2025; Dai et al., 2025). Among various structure understanding and generation tasks, Text-to-SQL (Tan et al., 2024; Lee et al., 2025; Dai et al., 2025) has emerged as a representative benchmark for evaluating the reasoning and compositional capabilities of LLMs. This task requires large language models to generate executable SQL queries from natural language questions, bridging unstructured human intent and structured database operations.

Recently, a variety of methods have been proposed to enhance the Text-to-SQL capabilities of LLMs. These approaches can be broadly categorized into two paradigms: prompt-based methods (Pourreza & Rafiei, 2023; Wang et al., 2025; Qin et al., 2025; Gao et al., 2023; Wu et al., 2025) and fine-tuning methods (Eyal et al., 2023; Li et al., 2024; Pourreza & Rafiei, 2024; Liu et al., 2025). Prompt-based methods mainly focus on designing effective instructions, demonstrations, or reasoning chains that guide LLMs to generate correct SQL queries. Few-shot learning (Brown et al., 2020) is a common prompt-based method, where LLMs are guided by a small set of annotated examples to adapt to the Text-to-SQL task. DIN-SQL (Pourreza & Rafiei, 2023) exemplifies this paradigm by decomposing the SQL generation task into sub-problems and incorporating their intermediate solutions into prompts for LLMs. Another prompt-based approach, MCP (Qin et al., 2025) extends the prompt-based paradigm by introducing a Multitask Collaboration Prompting strategy, which leverages multiple SQL-related tasks to mitigate hallucinations and further enhance Text-to-SQL performance. These methods are lightweight and require no parameter updates, making them flexible and cost-efficient. However, their performance is heavily sensitive to prompt quality and may lack robustness across different databases. In contrast, fine-tuning methods adapt LLMs to the Text-to-SQL task through supervised training (Luo et al., 2024), reinforcement learning (Zhang et al.,

2025a), or instruction tuning (Zhang et al., 2025b). For example, CodeS (Li et al., 2024) introduces an open-source series of pre-trained models with incremental SQL-centric pre-training and task-specific fine-tuning, achieving good performance with significantly smaller parameter sizes. QPL-SQL (Eyal et al., 2023) introduces the Query Plan Language (QPL) and augments the Spider dataset with it to train models. Unlike CodeS (Li et al., 2024) and QPL-SQL (Eyal et al., 2023), which focus on training models to reason, STaR-SQL (He et al., 2025) instead constructs task-specific data for verifier training. These fine-tuned models generally achieve higher performance and better generalization, but often require the additional synthesis of large-scale task-specific data. Such reliance on high-quality prompts and costly task-specific data highlights the necessity of developing more efficient and adaptable solutions.

Inspired by the fact that when humans learn a new concept, forward understanding of the solution facilitates initial knowledge acquisition, while reverse reasoning from the answer back to the question further consolidates comprehension, we design Progressive Reverse Understanding (**PRU**) for Text-to-SQL generation. We emphasize the utilization of progressive reverse understanding as a central mechanism to strengthen semantic alignment between natural language and structured SQL. Specifically, our method first leverages reverse data (SQL-to-Text), training the model to construct natural language questions from structured SQL queries, thereby fostering a deeper understanding of query structures and their semantic implications. After this initial stage, we progressively introduce forward data (Text-to-SQL) while retaining a smaller portion of reverse data to maintain bidirectional alignment. As training advances, the proportion of reverse data is gradually reduced, allowing the model to focus more on forward generation while still benefiting from the reverse reasoning signal. This progressive scheduling not only mitigates error propagation but also provides a smoother learning curve, leading to stronger semantic grounding and improved syntactic correctness. To demonstrate the effectiveness of our method, we conduct extensive experiments on the widely used Text-to-SQL benchmark Spider (Yu et al., 2018). Experimental results show that our approach consistently outperforms existing baselines and achieves good performance in terms of exact match accuracy (EM) and execution accuracy (EX). Further analysis validates the critical role of progressive reverse understanding in driving performance, confirming that the integration of reverse reasoning is essential for both syntactic validity and semantic fidelity.

Our main contributions are as follows:

- We propose **Progressive Reverse Understanding (PRU)**, a novel bidirectional training paradigm that integrates forward generation with reverse construction for Text-to-SQL.

- We design a progressive scheduling strategy that first emphasizes reverse reasoning to strengthen structural comprehension, and then gradually shifts towards forward generation for robust semantic alignment.

- Extensive experiments and in-depth analyses demonstrate the effectiveness of our proposed PRU framework, showing consistent improvements in both syntactic validity and semantic fidelity on the Spider benchmark.

## 2 RELATED WORK

Text-to-SQL task (Zhu et al., 2024; Rajkumar et al., 2022) focuses on generating executable SQL queries from natural language questions, bridging unstructured human intent and structured database operations. Most existing Text-to-SQL approaches are built upon large language models (LLMs) (Touvron et al., 2023; OpenAI, 2024), owing to their superior capabilities in natural language understanding (Radhakrishnan et al., 2023; Cheng et al., 2024) and generation (Zhong & Litman, 2025; Kim & Kim, 2025). These approaches can be categorized into two paradigms: prompt-based methods (Pourreza & Rafiei, 2023; Wang et al., 2025; Qin et al., 2025; Gao et al., 2023; Wu et al., 2025) and fine-tuning methods (Eyal et al., 2023; Li et al., 2024; Pourreza & Rafiei, 2024; Liu et al., 2025).

**Prompt-based Methods** Prompt-based Text-to-SQL methods (Pourreza & Rafiei, 2023; Wang et al., 2025; Qin et al., 2025; Gao et al., 2023; Wu et al., 2025) rely on carefully crafted instructions, demonstrations, or reasoning chains to steer LLMs toward generating accurate SQL queries. Few-shot learning (Brown et al., 2020), the most direct form of prompt-based methods, aims to adapt models to the task by selecting diverse examples. Building on this direction, UCS-SQL (Wu

et al., 2025) enhances example selection by jointly considering SQL skeletons and question expressions, thereby providing more targeted guidance for SQL generation. DIN-SQL (Pourreza & Rafiei, 2023) follows a similar paradigm by decomposing the SQL generation task into sub-problems and incorporating their intermediate solutions into prompts for LLMs. MCP (Qin et al., 2025) extends the prompt-based paradigm by introducing a Multitask Collaboration Prompting strategy, which leverages multiple SQL-related tasks to mitigate hallucinations and further enhance Text-to-SQL performance. Another representative approach, MAC-SQL (Wang et al., 2025), introduces a multi-agent collaborative prompting framework, where a decomposer agent with chain-of-thought reasoning coordinates with auxiliary agents that leverage external tools to further enhance Text-to-SQL performance. These methods are lightweight and require no parameter updates, making them flexible. However, their performance often heavily depends on prompt quality and may lack robustness across different databases.

**Fine-tuning Methods** Fine-tuning Text-to-SQL methods (Eyal et al., 2023; Li et al., 2024; Pourreza & Rafiei, 2024; Liu et al., 2025) adapt LLMs to the Text-to-SQL task through supervised training (Luo et al., 2024), reinforcement learning (Zhang et al., 2025a), or instruction tuning (Zhang et al., 2025b) with additional task-specific data. For instance, QPL-SQL (Eyal et al., 2023) augments the original dataset with Query Plan Language (QPL) and fine-tunes on the augmented data to improve Text-to-SQL performance. CodeS (Li et al., 2024) trains a suite of open-source models on a curated SQL-centric corpus constructed via data augmentation, achieving competitive performance with substantially fewer parameters. DTS-SQL (Pourreza & Rafiei, 2024) introduces a two-stage fine-tuning framework that first addresses schema linking and then performs SQL generation. STaR-SQL (He et al., 2025) trains a verifier utilizing augmented correct and incorrect solutions data, enhancing test-time verification and overall performance. Different from CodeS (Li et al., 2024), DTS-SQL (Pourreza & Rafiei, 2024), and STaR-SQL (He et al., 2025) which rely on supervised fine-tuning (SFT), CoT-DPO (Liu et al., 2025) synthesizes Chain-of-Thought (Wei et al., 2022) data and employs Direct Preference Optimization (DPO) (Rafailov et al., 2023) training, thereby further improving Text-to-SQL performance. These fine-tuned models generally achieve higher performance, but often require the additional synthesis of large-scale task-specific data.

## 3 METHODOLOGY

In this section, we first give a formulation of the Text-to-SQL task in §3.1 and then describe our method in detail. As shown in Figure 1, the framework of our method consists of two components, a reverse construction module §3.2 and a forward generation module §3.3.

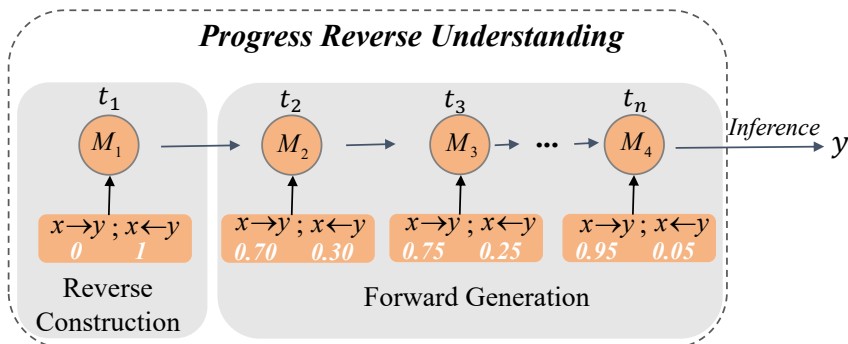

Figure 1: Illustration of our progress reverse understanding method with two modules: 1)Reverse Construction, which trains the model to construct natural language questions from structured SQL queries (SQL-to-Text), thereby facilitating a deeper understanding of query structures and their semantic implication. 2) Forward generation, which fine-tunes the model primarily on Text-to-SQL data while retaining a diminishing portion of reverse data, enables the model to focus more on forward generation (Text-to-SQL) while still benefiting from the reverse reasoning signal.

## 3.1 TASK FORMULATION

The goal of the Text-to-SQL task is to translate a natural language (NL) question into a corresponding SQL query that can be executed over a relational database. Formally, given a natural language input $x$ and a database schema $\mathcal{S}$, the model is required to generate a SQL query $y$ such that its execution result $\text{Exec}(y, \mathcal{S})$ correctly answers the question $x$.

Let $\mathcal{D} = \{(x_i, y_i, \mathcal{S}_i)\}_{i=1}^N$ denote the training dataset, where $x_i$ is a natural language utterance, $y_i$ is its corresponding SQL query, and $\mathcal{S}_i$ is the schema of the target database. The mapping function can be expressed as:

$$f_\theta : (x, \mathcal{S}) \to y, \tag{1}$$

where $f_\theta$ is a neural model parameterized by $\theta$.

The key challenge of this task lies in the compositional and structural nature of SQL: the model must not only capture the semantics of the input utterance but also align it with schema elements (tables, columns, values) in $\mathcal{S}$, while ensuring that the generated query $y$ is both syntactically valid and semantically faithful. Traditional fine-tuning based approaches rely on additional task-specific data on the forward mapping $x \to y$, which often suffers from high annotation cost. These motivate the need for a more efficient paradigm that can leverage existing data more effectively while reducing reliance on costly task-specific supervision.

## 3.2 REVERSE CONSTRUCTION

The reverse construction module is designed to enhance the model's structural understanding of SQL by training it to map SQL queries back into their corresponding natural language utterances. Formally, given a SQL query $y$ and its associated schema $\mathcal{S}$, the model is optimized to generate the original question $x$. The objective can be expressed as:

$$\mathcal{L}_{\text{reverse}} = - \sum_{(x,y,\mathcal{S}) \in \mathcal{D}} \log P_\theta(x \mid y, \mathcal{S}). \tag{2}$$

This reverse process mimics how humans consolidate knowledge by reasoning backward from answers to questions. Through reverse training, the model develops stronger awareness of SQL syntax and semantics, learns to align query structures with natural language meaning, and builds a foundation for subsequent forward generation. The reverse construction module thus provides essential structural grounding, which complements the forward generation process.

## 3.3 FORWARD GENERATION

The forward generation module addresses the standard Text-to-SQL task, where the model generates SQL queries from natural language utterances. Formally, given an input question $x$ and schema $\mathcal{S}$, the model predicts the corresponding SQL $y$:

$$\mathcal{L}_{\text{forward}} = - \sum_{(x,y,\mathcal{S}) \in \mathcal{D}} \log P_\theta(y \mid x, \mathcal{S}). \tag{3}$$

Unlike conventional forward-only training, our approach integrates the forward generation module with the reverse construction module through a **progressive training strategy**. The training procedure unfolds as follows:

- **Step 1:** Train the initial model $M_1$ solely on reverse data $(y, x)$, equipping the model with SQL comprehension and the ability to reason backward.

- **Step 2:** Starting from $M_1$, fine-tune the model using a mixture of 70% forward data $(x, y)$ and 30% reverse data $(y, x)$, obtaining model $M_2$.

- **Step 3:** Starting from $M_2$, fine-tune on 75% forward and 25% reverse data, yielding model $M_3$.

- **Subsequent Steps:** Continue this process, progressively increasing the proportion of forward data while decreasing the reverse proportion, until the ratio reaches 95% forward and 5% reverse. The resulting model $M_n$ is then adopted for downstream Text-to-SQL tasks.

This progressive design ensures that the model first builds structural understanding through reverse learning, and then gradually adapts to forward generation while retaining a small portion of reverse signals. Such bidirectional alignment enhances semantic consistency between natural language and SQL, leading to improved syntactic validity and execution accuracy.

## 3.4 INFERENCE

At inference time, we employ the final model $M_n$ obtained through the progressive training strategy. Given a natural language question $x$ and its corresponding database schema $\mathcal{S}$, the model directly generates a SQL query $\hat{y}$ using the forward generation module:

$$\hat{y} = \arg\max_y M_n(y \mid x, \mathcal{S}). \tag{4}$$

Different from the training stage, inference does not require the reverse construction module, as its role is primarily to guide the model during learning. However, the knowledge acquired from reverse reasoning remains implicitly encoded in the model parameters, enabling more accurate generation.

Since $M_n$ has been progressively fine-tuned with a decreasing proportion of reverse data, it preserves structural awareness while being highly optimized for forward generation. This design ensures that the generated SQL queries are not only syntactically valid but also semantically faithful to the input question, thereby improving execution accuracy.

## 4 EXPERIMENTS

### 4.1 DATASETS AND EVALUATION

**Datasets** To demonstrate the effectiveness of our proposed approach, we conduct experiments on the standard Text-to-SQL dataset Spider (Yu et al., 2018). The Spider dataset contains 10,181 natural language questions paired with 5,693 distinct complex SQL queries spanning 200 databases across 138 domains, each involving multiple tables. Following the standard evaluation protocol, the dataset is split into 8,659 training samples from 146 databases and 1,034 development samples from 20 databases, with no database overlap between the two sets. SQL queries are assigned to four difficulty levels according to factors such as the number of SQL keywords, the existence of nested subqueries, and the complexity of column selection and aggregation. This dataset serves as a benchmark for testing the ability of text-to-SQL models to generalize to complex queries over unseen database schemas.

**Evaluation** Following existing methods (Pourreza & Rafiei, 2023; Wang et al., 2025; Li et al., 2024; Pourreza & Rafiei, 2024), we use exact match accuracy (EM) and execution accuracy (EX) to measure the performance of our method. Specifically, the exact-set-match accuracy (EM) evaluates predictions by treating each SQL clause as a set and comparing it with the corresponding clause in the reference query. A predicted query is considered correct only if all components exactly match the ground truth, and values are not taken into account. In contrast, execution accuracy (EX) measures agreement between the execution results of the predicted and reference SQL queries on database instances. Since multiple SQL queries can yield the same result for a given question, EX offers a more reliable estimate of model performance than EM, which only checks against a single reference query. Our experiment results on the Spider dataset are evaluated using the released official evaluation code [1].

### 4.2 BASELINES

To evaluate the performance of our proposed method PRU, we compare it with several existing baseline methods. These baseline methods provide a benchmark for evaluating the relative performance of our proposed approach. We compare our method with 11 baselines, categorized into two groups.

---

[1]https://github.com/taoyds/spider

**Prompt-based Methods** : We first include the **Few-shot** method, the most direct form of prompt-based Text-to-SQL, as a fundamental baseline for comparison. **DIN-SQL** (Pourreza & Rafiei, 2023) exemplifies this paradigm by decomposing the SQL generation task into sub-problems and incorporating their intermediate solutions into prompts for LLMs. MCP (Qin et al., 2025) extends the prompt-based paradigm by introducing a Multitask Collaboration Prompting strategy, which leverages multiple SQL-related tasks to mitigate hallucinations and further enhance Text-to-SQL performance.

**Fine-tuning Methods** **Vanilla** fine-tuning methods adapt LLMs to the Text-to-SQL task using Text-to-SQL data. This baseline provides a straightforward reference point to evaluate the benefits of more advanced fine-tuning strategies. **CodeS** (Li et al., 2024) trains a series of open-source models on a curated SQL-centric corpus built with bi-directional data augmentation, achieving strong performance with substantially smaller parameter sizes. QPL-SQL (Eyal et al., 2023) introduces the Query Plan Language (QPL) and augments the Spider dataset with it to train models. Different from CodeS (Li et al., 2024) and QPL-SQL (Eyal et al., 2023), which focus on training models to reason, STaR-SQL He et al. (2025) instead constructs task-specific data for verifier training.

### 4.3 IMPLEMENTATION DETAILS

We implement our method based on the Llama-3.1-8B-Instruct (Meta-AI, 2024) model. All progressive training steps are carried out using the LoRA fine-tuning strategy, which enables efficient adaptation of large language models with limited computational resources. We train the reverse construction stage for 3 epochs, and all subsequent stages for 1 epoch. The main hyper-parameters are set as follows: the per-device training batch size is 1, with gradient accumulation steps of 8 to achieve an effective batch size of 8. We use a learning rate of $1.0 \times 10^{-5}$ and adopt a cosine learning rate scheduler with a warmup ratio of 0.1. These settings are applied consistently across all stages of progressive training, from the initial reverse pre-training to the final mixed fine-tuning stage.

## 5 RESULTS AND ANALYSIS

### 5.1 MAIN RESULTS

To evaluate the effectiveness of our method, we conduct experiments on the Spider (Yu et al., 2018) dataset, and the results are presented in Table 1. Results show that our proposed method, PRU-SQL, achieves the best overall performance with an execution accuracy (EX) of 75.8% and an exact-set-match accuracy (EM) of 70.6%, outperforming all prompt-based and fine-tuning baselines. More importantly, our PRU-SQL achieves a substantial improvement in exact-set-match accuracy (EM), surpassing the baselines by a clear margin. Specifically, the strongest baseline, STaR-SQL (He et al., 2025), obtains 64.9% EM, PRU-SQL pushes this further to 70.6%, highlighting its advantage in producing syntactically precise and semantically faithful SQL queries rather than relying solely on execution equivalence.

**Versus Prompt-based Methods.** For the Prompt-Based methods, we first examine the Few-shot setting. To provide a more comprehensive comparison, we report results not only from open-source models (e.g., Llama-3.1-8B-Instruct (Meta-AI, 2024)), but also from several closed-source models, including GPT-4 (OpenAI, 2024), CodeX Cushman, and CodeX Davinci (OpenAI, 2021). The results in Table 1 demonstrate that Few-shot learning shows highly variable performance across different LLMs, with Llama-3.1-8B-Instruct reaching only 55.0% EX, while GPT-4 achieves a stronger 67.4% EX and 54.3% EM. DIN-SQL (Pourreza & Rafiei, 2023) enhances standard prompting by decomposing SQL generation into sub-problems, achieving 74.2% EX and 60.1% EM with Llama-3.1-8B-Instruct, yet it still falls short of our approach. MCP (Qin et al., 2025) leverages multitask collaboration and reaches a strong 75.0% EX; however, our method PRU continues to outperform it.

**Versus Fine-tuning Baselines.** As demonstrated by the experimental results, our PRU consistently achieves the best performance among all fine-tuning methods. It is worth noting that our Vanilla approach yields moderate performance (69.8% EX, 58.6% EM), surpassing the performance of QPL-SQL (Eyal et al., 2023) (64.5% EX, 54.3% EM), even though QPL-SQL introduces the Query Plan Language. We attribute this improvement to the stronger instruction we adopt during

Table 1: Evaluation results on Spider dataset. Vanilla denotes the traditional supervised fine-tuning training method on same large language model.

| Types | Methods | LLMs | EX | EM |
|---|---|---|---|---|
| Prompt-Based | Few-shot | Llama-3.1-8B-Instruct | 55.0 | 34.2 |
| | | GPT-4 | 67.4 | 54.3 |
| | | CodeX Cushman | 43.1 | 30.9 |
| | | CodeX Davinci | 61.5 | 50.2 |
| | DIN-SQL (Pourreza & Rafiei, 2023) | Llama-3.1-8B-Instruct | 45.2 | 26.5 |
| | | GPT-4 | 74.2 | 60.1 |
| | MCP (Qin et al., 2025) | Llama-3-8B | 75.0 | – |
| Fine-Tuning | Vanilla | Llama-3.1-8B-Instruct | 69.8 | 58.6 |
| | QPL-SQL (Eyal et al., 2023) | Llama-3.1-8B-Instruct | 64.5 | 54.3 |
| | CodeS (Li et al., 2024) | StarCoder | 69.8 | – |
| | STaR-SQL (He et al., 2025) | Llama-3.1-8B-Instruct | 75.0 | 64.9 |
| | **PRU-SQL (Ours)** | **Llama-3.1-8B-Instruct** | **75.8** | **70.6** |

fine-tuning. Furthermore, our PRU-SQL surpasses CodeS (Li et al., 2024), despite the latter being fine-tuned on specialized code LLMs. Compared with STaR-SQL (He et al., 2025), our PRU-SQL achieves comparable execution accuracy (EX), but delivers a substantially higher exact match (EM), demonstrating that our method not only executes correctly but also generates more precise and semantically faithful SQL queries.

## 5.2 ANALYSIS

This section provides a comprehensive analysis of why and how the proposed Progressive Reverse Understanding (PRU) improves Text-to-SQL performance.

**Ablation Studies** As demonstrated in Methodology (Section 3), our proposed PRU method consists of two components. To investigate the impact of each component in detail, we conduct a series of ablation experiments using Llama-3.1-8B-Instructs (Meta-AI, 2024) on the Spider dataset (Yu et al., 2018). Specifically, we consider the following settings: (i) No Reverse Pre-training, where the reverse pre-training stage ($M_1$) is removed and training is performed only on $(x, y)$ pairs, in order to test the importance of early reverse exposure; (ii) No Progressive Schedule, where the reverse-to-forward mixing ratio is fixed ($0.3$ to $0.7$) throughout training, to evaluate the role of gradually shifting toward forward generation; (iii) No Residual Reverse Signal, where the final stage uses $100\%$ $(x, y)$ pairs, aiming to assess whether a small amount of reverse signal is necessary for stability; and (iv) Reverse-Only or Forward-Only, where training is restricted to a single direction, to verify the necessity of incorporating both.

The results in Table 2 show that removing the reverse pre-training stage leads to a substantial performance drop (EX 71.7, EM 66.9), confirming the critical role of early reverse exposure in establishing structural grounding. When the progressive scheduling is disabled and a constant mixing ratio is applied, performance decreases to EX 73.3 and EM 68.1, which highlights the importance of gradually shifting toward forward generation to provide an effective curriculum. In contrast, removing the residual 5% reverse signal at the final stage results in a smaller degradation (EX 74.4, EM 69.7), yet it

Table 2: Ablation studies performance on spider dataset with Llama-3.1-8B-Instruct.

| Methods | EX | EM |
|---|---|---|
| **Full model** | **75.8** | **70.6** |
| w/o Reverse Pre-training | 71.7 | 66.9 |
| w/o Progressive Schedule | 73.3 | 68.1 |
| w/o Residual Reverse Signal | 74.4 | 69.7 |
| Reverse-Only | 64.2 | 60.3 |
| Forward-Only | 68.3 | 57.5 |

still demonstrates that maintaining a persistent bidirectional signal stabilizes the model. Finally, both reverse-only and forward-only training settings yield the weakest results (EX $\approx$ 64.2, EM $\approx$

60.3), which validates that the synergy of both directions is indispensable for optimal Text-to-SQL performance.

$M_1$ **Duration Sensitivity**   To investigate how long the reverse-only pre-training stage ($M_1$) should last in order to achieve the best trade-off between structural grounding and forward generation, we conduct a $M_1$ Duration Sensitivity analysis. Specifically, we fix the overall training budget to $E_{\text{total}}$ epochs and vary the reverse-only pre-training duration $E_{M_1} \in \{0, 1, 2, 3, 5, 8\}$. The subsequent stage keeps the same reverse-to-forward schedule across settings. We report EX accuracy at checkpoints to identify the minimal effective $E_{M_1}$ and the diminishing-returns region.

Table 3: Effect of varying the $M_1$ duration (in epochs) on model performance on the Spider dataset. The best result is highlighted in bold.

| $E_{M_1}$ (epochs) | 0 | 1 | 2 | **3** | 5 | 8 |
|---|---|---|---|---|---|---|
| EX (%) | 71.7 | 72.4 | 74.1 | **75.8** | 74.9 | 74.0 |
| EM (%) | 66.9 | 67.0 | 68.9 | **70.6** | 69.8 | 69.0 |

The analysis results are shown in Table 3. Intuitively, too short an $M_1$ may fail to provide sufficient structural exposure, while too long an $M_1$ may overemphasize the reverse direction and harm subsequent forward generalization. By systematically varying the number of epochs devoted to $M_1$ , we aim to identify the minimal effective duration and the diminishing-returns region, as well as to better understand the role of early reverse exposure in our proposed method PRU. The results in Table 3 reveal a clear trend: extending $M_1$ from 0 to 3 epochs consistently improves both EX and EM, demonstrating the effectiveness of early reverse exposure for structural grounding. The best performance is achieved when $E_{M_1} = 3$, yielding 75.8% EX and 70.6% EM. Beyond this point, however, further prolonging $M_1$ (e.g., to 5 or 8 epochs) leads to a slight decline in performance, suggesting that excessive reverse training may hinder the model's ability to adapt to the forward generation objective. These findings indicate that a moderate amount of reverse pre-training strikes the optimal balance between structural alignment and forward generalization, thereby validating the necessity of carefully tuning $M_1$ duration in our proposed PRU.

**Curriculum Shape & Turning Point**   To analyze how the shape of the reverse-to-forward mixing schedule and the choice of its turning point influence model performance, we design a Curriculum Shape & Turning Point experiment. Our hypothesis is that a gradual curriculum facilitates smoother structural-to-generative adaptation, while abrupt or poorly timed transitions may either underutilize reverse signals or delay forward generalization. We keep the overall training budget and $E_{M_1}$ fixed, and vary the reverse-to-forward curriculum along two dimensions: 1) Schedule Shape. Linear Decay: the reverse ratio decreases linearly from $r_{\max}$ to $r_{\text{res}}$. 2) Turning Point. We control the step at which the reverse ratio begins to increase, comparing 30%, 50%, 70%, and 80% of $M_2$ schedules. All other training settings, including data, optimizer hyperparameters, and decoding configurations, remain identical.

From the results shown in Figure 2, we observe that the choice of the turning point in the reverse-to-forward curriculum has a significant impact on model performance. When the turning point is set too early (30% or 50%), both EX and EM scores decrease, indicating that the model fails to sufficiently benefit from the reverse signals and thus struggles to generalize effectively in the forward phase. In contrast, when the turning point is delayed to 70%, performance begins to recover, suggesting that maintaining a higher proportion of reverse data for a longer period allows the model to better leverage structural supervision before shifting toward generative training. This finding directly motivates our design choice: we initialize $M_1$ with 100% reverse data, and set $M_2$ to start from 30% reverse and 70% forward data.

**Residual Reverse Ratio Sweep**   This experiment aims to investigate the role of the residual reverse ratio $r_{\text{res}}$ in the final stage of training. Intuitively, leaving a small proportion of reverse signals may stabilize the alignment between reverse and forward generation, while removing them entirely ($r_{\text{res}} = 0$) could cause the model to overfit to forward-only objectives. Conversely, keeping too much reverse signal might hinder forward adaptation. Therefore, we seek to determine the optimal residual ratio that balances persistent structural grounding with forward generalization. We fix the

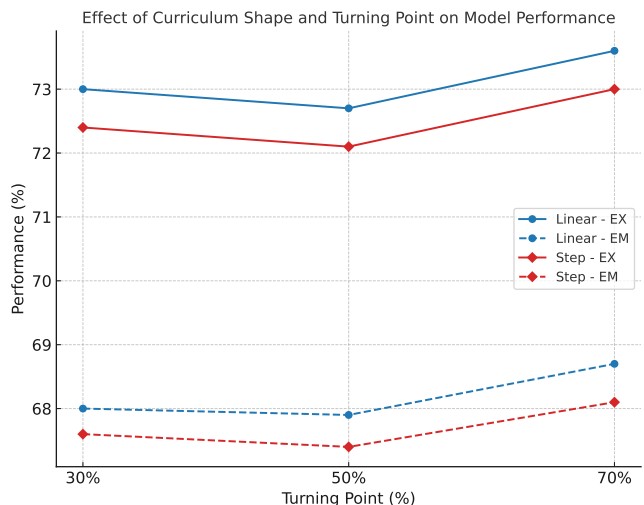

Figure 2: Comparison of curriculum shapes and turning points.

total training budget, the $M_1$ duration ($E_{M_1}$), and the reverse-to-forward curriculum schedule (shape and turning point). At the final stage of $M_n$, we vary the residual reverse ratio $r_{\text{res}}$ across a set of values: $r_{\text{res}} \in \{0\%, 1\%, 5\%, 10\%, 20\%\}$

The results in Table 4 demonstrate that leaving a small but non-zero residual reverse ratio is essential for achieving optimal performance. When $r_{\text{res}} = 0\%$, both EX and EM drop significantly, confirming that completely removing reverse signals at the final stage leads to unstable training and weaker structural grounding. Increasing the residual ratio to 1% improves performance slightly, while 5% achieves the best overall results (75.8% EX and 70.6% EM), striking the right balance between persistent bidirectional alignment and forward generalization. Further increasing the ratio to 10% or 20% yields diminishing or negative returns, as excessive reverse signals interfere with the model's ability to adapt to forward generation. These findings highlight the stabilizing effect of a small residual reverse signal and suggest that around 5% is an optimal choice for maintaining robustness without hindering forward performance.

Table 4: Effect of varying the residual reverse ratio $r_{\text{res}}$ on model performance (Spider dev set). A small but non-zero residual (5%) yields the best results.

| $r_{\text{res}}$ | EX (%) | EM (%) |
|---|---|---|
| 0% | 73.5 | 68.2 |
| 1% | 74.6 | 69.5 |
| 5% | **75.8** | **70.6** |
| 10% | 75.1 | 70.0 |
| 20% | 74.0 | 69.1 |

## 6 CONCLUSION

In this paper, we propose a **Progressive Reverse Understanding (PRU)** method for Text-to-SQL generation, a novel training paradigm that incorporates reverse construction into a progressive forward learning process. By first training the model to reason backward from SQL to natural language and then gradually shifting the focus to forward generation with a decreasing proportion of reverse data, PRU effectively strengthens structural comprehension, enhances semantic alignment, and mitigates error propagation. Extensive experiments on benchmark datasets demonstrate that PRU achieves good performance. Furthermore, ablation studies and detailed analyses verify the effectiveness of each component in our proposed framework, highlighting the contributions of individual modules.

ETHICS STATEMENT

This research does not involve human subjects, personally identifiable information, or sensitive data. All datasets used in this work are publicly available and widely used in the community. We have followed responsible research practices and ensured that no private or proprietary data was included. The proposed methodology is intended for academic research purposes and does not pose foreseeable risks of misuse. We believe our study complies with the ICLR Code of Ethics.

REPRODUCIBILITY STATEMENT

To ensure the reproducibility of our work, we have included the complete implementation of our proposed method in the supplementary materials. The released code is accompanied by a detailed `README.md` file, which provides step-by-step instructions and execution commands for reproducing our experiments. This includes dataset preprocessing, model training, and evaluation procedures. With these resources, other researchers can readily verify and build upon our results.

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

## A    THE USE OF LARGE LANGUAGE MODELS (LLMS)

In this work, Large Language Models (LLMs) are employed solely as auxiliary tools for language polish. Specifically, we make use of LLMs to polish the wording of our manuscript without altering the underlying technical content or experimental results. For this purpose, we adopt a simple prompt, `"please proof this demonstration slightly."`, to improve readability and stylistic consistency. No LLM-generated content related to methodology design, implementation, or experimental results is included in this paper.

