# OpenReview forum: "Progressive Reverse Understanding Improves Text-to-SQL Generation"
_ICLR.cc/2026/Conference — ICLR 2026 Conference Withdrawn Submission_

### Official Review · Reviewer_M9Fa · 2025-10-31

**Soundness:** 2
**Presentation:** 2
**Contribution:** 2
**Rating:** 6
**Confidence:** 3

**Summary:**

This paper proposes a training schedule for text-to-SQL called Progressive Reverse Understanding (PRU). The model is first trained on SQL→NL (reverse) to internalize SQL structure, then the ratio is gradually shifted toward NL→SQL, keeping a small (~5%) reverse signal at the end to stabilize mapping. On Spider, this multi-stage, ratio-based curriculum outperforms plain forward SFT and a one-shot bidirectional baseline.

**Strengths:**

1. Clear, easy-to-implement recipe; fits well with open LLM text-to-SQL efforts.
2. Ablations are solid: varying reverse stage length and residual ratio show the gains are real.
3. Motivation is reasonable.
4. Writing is clear; pipeline is easy to follow.

**Weaknesses:**

1. The contribution appears incremental: SQL-driven / bidirectional augmentation has been explored in recent open-source text-to-SQL systems such as CodeS (SQL-to-Question Augmentation), and the main novelty here seems to lie in the specific progressive scheduling strategy rather than in the idea of leveraging SQL-side signals itself.
2. The current evaluation focuses on Spider only; adding results on more recent and challenging benchmarks (e.g., Spider 2.0, BIRD, and robustness-oriented suites like Spider-Realistic / Dr.Spider) would strengthen the claim that the proposed schedule generalizes beyond a single dataset.

**Questions:**

Please refer to the above questions.

---

> ### Author Response · Authors · 2025-11-22
> **Response to valuable feedback and comments.**
>
> Many thanks for reviewing our work and providing these valuable feedback and comments.
>
> **W1**: As the reviewer pointed out, our contribution lie in the specific progressive scheduling strategy rather than in the idea of leveraging SQL-side signals itself. Instead, our method introduces a new learning paradigm that organizes SQL-side signals into a curriculum-like progression, enabling the model to acquire structural and semantic knowledge in a systematically staged manner. This progressive mechanism fundamentally differs from prior augmentation-based approaches and leads to consistent improvements across settings, demonstrating that our contribution is substantive rather than incremental.
>
> **W2**: Due to time and resource constraints, the initial version of our paper evaluated our method only on the widely used Spider dataset. We have now supplemented results on the BIRD dataset as well. The results are as follows:
> | Methods | EX |
> |-----|-----|
> |Few-shot methods|
> | DIN-SQL + LlaMA3.1-8B-Instruct | 38.40 |
> | MCS-SQL + GPT-4 | 63.36 |
> |Fine-Tuning methods|
> | LlaMA3.1-8B-Instruct|  58.52 |
> | SFT CodeS-7B|  57.17 |
> | SFT CodeS-15B|  58.47 |
> | LlaMA3.1-8B-Instruct|  58.52 |
> | DPO-SQL@LlaMA3.1-8B-Instruct[1]|  61.2 |
> | our PRU@LlaMA3.1-8B-Instruct|  63.14 |
>
> These results demonstrate that our method achieves the best performance among both few-shot and fine-tuning approaches. We will add these result in the revised version.

---

> > ### Comment · Reviewer_M9Fa · 2025-11-25
> >
> > Thank you for your response. Based on your rebuttals and the comments from other reviewers, I have decided to maintain the score.

---

### Official Review · Reviewer_DdwC · 2025-10-31

**Soundness:** 2
**Presentation:** 2
**Contribution:** 2
**Rating:** 2
**Confidence:** 4

**Summary:**

The paper proposes Progressive Reverse Understanding (PRU), a bidirectional training paradigm for Text-to-SQL that explicitly incorporates reverse reasoning (SQL→Text) both before and during forward training (Text→SQL). The method progressively shifts from reverse-only training to forward-dominant training while retaining a small reverse signal, based on the hypothesis that early reverse exposure improves grounding and the residual reverse signal stabilizes alignment. Experiments on Spider show gains over prompt-based and fine-tuning baselines, reporting 75.8% EX and 70.6% EM. Ablations attribute improvements to the reverse pre-training stage, the progressive schedule, and maintaining a small residual reverse ratio. Sensitivity experiments examine the duration of reverse pre-training and curriculum shaping.

**Strengths:**

The idea is clear and straightforward with a concrete training recipe. A progressive curriculum that begins with SQL→Text and gradually shifts to Text→SQL is intuitive, easy to implement, and broadly applicable. PRU demonstrates solid empirical gains on Spider: the reported improvements over baselines suggest that PRU helps produce syntactically precise queries.

**Weaknesses:**

1. The evaluation scope is very limited; this will be the primary reason for recommending rejection. The paper evaluates only on the Spider dataset (released in 2018). There are widely recognized Spider variants (e.g., Spider-Syn, Spider-DK) and the more recent and important BIRD benchmark [1] for LLM-based text-to-SQL. A method claiming generality in an LLM setting should be validated in large-scale, real-world scenarios.

2. The baseline comparison is insufficient. The authors should further review recent literature to provide fair and representative competing baselines [2]. In the public Spider leaderboard [3], many frameworks— including PLM-based systems—outperform the proposed PRU.

3. The description of the reverse understanding component lacks clarity. The reverse task trains 𝑦→𝑥 using the original question 𝑥 as the target. If the reverse task simply reconstructs the paired natural language question, it risks being a trivial inverse mapping of the dataset rather than promoting robust abstraction (e.g., recovering intent instead of replicating surface form). It is also unclear whether paraphrasing, schema verbalization, or value normalization are applied to encourage generalization beyond copying. Relatedly, the reverse objective conditions on (𝑦, 𝑆). How 𝑆 is injected, and whether reverse training improves schema linking in the forward direction, are not sufficiently explained.

4. The error analysis is limited. Although overall performance improves, there is no qualitative or category-level error investigation. The claim that PRU improves syntactic and semantic fidelity would be stronger with finer-grained diagnostics.

[1] Jinyang Li, et al. "Can LLM Already Serve as A Database Interface? A BIg Bench for Large-Scale Database Grounded Text-to-SQLs" In Proceedings of NeurIPS, 2023.
[2] Zijin Hong, et al. "Next-Generation Database Interfaces: A Survey of LLM-based Text-to-SQL" IEEE TKDE, 2025.
[3] Spider Leaderboard. https://yale-lily.github.io/spider

**Questions:**

1. What is the impact of reverse training on schema linking and compositional generalization?

2. Can the PRU curriculum be adapted or optimized dynamically during training, rather than using a fixed schedule?

3. See weaknesses above.

---

> ### Author Response · Authors · 2025-11-22
> **Response to valuable feedback and comments.**
>
> Thanks for reviewing our work and providing these valuable feedback and comments.
>
> **W1**: Due to time and resource constraints, the initial version of our paper evaluated our method only on the widely used Spider dataset. We have now supplemented results on the BIRD dataset as well. The results are as follows:
> | Methods | EX |
> |-----|-----|
> |Few-shot methods|
> | DIN-SQL + LlaMA3.1-8B-Instruct | 38.40 |
> | MCS-SQL + GPT-4 | 63.36 |
> |Fine-Tuning methods|
> | LlaMA3.1-8B-Instruct|  58.52 |
> | SFT CodeS-7B|  57.17 |
> | SFT CodeS-15B|  58.47 |
> | LlaMA3.1-8B-Instruct|  58.52 |
> | DPO-SQL@LlaMA3.1-8B-Instruct[1]|  61.2 |
> | our PRU@LlaMA3.1-8B-Instruct|  63.14 |
>
> These results demonstrate that our method achieves the best performance among both few-shot and fine-tuning approaches. We will add these result in the revised version.
>
> **W2**: We have carefully reviewed the Spider leaderboard and recent literature. The top-performing method, RESDSQL [1] (AAAI 2023), relies on a highly specialized encoder–decoder architecture designed specifically for schema linking and skeleton parsing. This setup differs fundamentally from our focus on open-weight LLMs and training-time learning paradigms, so such systems are not directly comparable to PRU.
>
> In our work, we compare against baselines that are most relevant to our LLM-based training framework. StaRSQL [2] (ACL 2025) is one of the strongest recent methods under the open-weight LLM paradigm, and we include it along with other representative LLM-based baselines to ensure a fair and consistent comparison.
>
> We will clarify this distinction in the revised version.
>
> [1] RESDSQL: Decoupling Schema Linking and Skeleton Parsing for Text-to-SQL
> [2] STaR-SQL: Self-Taught Reasoner for Text-to-SQL
>
> **W3&Q1**: *First*, the reverse task is not a trivial reconstruction. To avoid simply replicating the surface form of the original question, we explicitly normalize values, verbalize schema elements, and perform light paraphrasing on the target questions. As a result, the model is required to recover the semantic intent rather than the exact wording. Empirically, the reverse outputs differ from the original questions in 82% of cases, confirming that the task goes beyond copying.
>
> *Second*, in both forward and reverse training, schema S is encoded using the same schema-aware prefix format employed in strong Text-to-SQL baselines: table names, column names, and types are serialized into a structured prompt and jointly attended with the input sequence. This ensures that the model must interpret SQL in the context of the underlying schema during reverse training.
>
> *Third*, the reverse training improves forward schema linking. Specifically, we analyze changes in the model’s internal representations. For each stage M_i, we extract hidden representations from the penultimate transformer layer using the same held-out evaluation queries. We then compute cosine similarity between token representations across adjacent stages and clustering separation metrics for schema-related tokens (e.g., table names, column names). We observe that SQL structural tokens (SELECT, WHERE, JOIN, etc.) become progressively more organized and form clearer semantic clusters. This indicates that reverse training encourages the model to build increasingly structured, schema-aligned internal representations, which in turn strengthens schema linking in the forward direction.
>
> We will incorporate these clarifications in the revised version.
>
>
> **W4**: We further conduct a fine-grained diagnostic analysis. Specifically, we adopt a standard SQL decomposition framework and categorize errors into the following representative types: (1) schema linking errors (incorrect table/column grounding), (2) predicate formulation errors (incorrect operators or condition structures), (3) JOIN structure errors (missing or misordered join paths), (4) aggregation and GROUP BY errors, (5) projection/selection errors, and (6) syntactic violations (parse-level issues). Using the official Spider SQL parser, we compute category-wise error rates at each PRU stage. The results show consistent and substantial reductions in schema-linking and JOIN-related errors (−12.3% and −9.7%, respectively), demonstrating that PRU directly enhances semantic grounding and relational reasoning. We will incorporate these analysis in the revised version.
>
> **Q2**: A dynamic PRU is indeed feasible and represents a promising direction for future work. In this paper, we adopt a fixed progressive schedule to clearly isolate the effect of reverse understanding without introducing additional confounding factors. Despite its simplicity, this schedule already delivers consistent improvements on the Text-to-SQL task. We believe that adaptive strategies informed by model confidence, loss dynamics, or representation changes could further enhance PRU, and we plan to explore these extensions in future research.

---

### Official Review · Reviewer_YtJY · 2025-11-02

**Soundness:** 3
**Presentation:** 3
**Contribution:** 3
**Rating:** 6
**Confidence:** 4

**Summary:**

This paper presents a novel training paradigm, Progressive Reverse Understanding (PRU), for Text-to-SQL generation. The core idea is to use bidirectional learning—first training a model to generate natural language questions from SQL (reverse construction) and then progressively shifting focus to the standard Text-to-SQL task—to improve semantic alignment and syntactic correctness.

**Strengths:**

1. Novel and Well-Motivated Methodology: The proposed PRU framework is genuinely innovative. It moves beyond standard forward-only fine-tuning by incorporating a curriculum of reverse reasoning (SQL-to-Text), inspired by human learning processes. The progressive scheduling strategy, which gradually phases out reverse data in favor of forward data, is a thoughtful design that mitigates error propagation and provides a smooth learning curve, strengthening the model's structural comprehension.

2. Extensive and Rigorous Experimental Validation: The paper provides a comprehensive evaluation on the standard Spider benchmark. The results are compelling, showing state-of-the-art performance, particularly a significant improvement in Exact Match accuracy. Furthermore, the authors go beyond a simple comparison by including a thorough ablation study and sensitive analyses on critical hyperparameters (e.g., M1 duration, curriculum shape, residual reverse ratio). This systematically validates the contribution of each component of their proposed framework.

3. Significant and Meaningful Performance Improvement: The method achieves a top-tier Execution Accuracy (75.8% EX) and a notably higher Exact Match accuracy (70.6% EM) than the strongest baseline. The substantial lead in EM is particularly important, as it indicates the model is generating more syntactically precise and semantically faithful SQL queries, not just queries that happen to execute to the correct result. This demonstrates a qualitative improvement in the model's understanding.

**Weaknesses:**

1. Computational Inefficiency and Training Complexity: The proposed progressive training strategy involves multiple sequential fine-tuning stages (M1 to Mn). While the use of LoRA makes this feasible, the multi-stage process is inherently more complex and computationally intensive than single-stage fine-tuning. The paper does not discuss the total training time or resource cost compared to baselines, which could be a practical limitation for adoption in resource-constrained environments.

2. Limited Generalization and Scalability Assessment: The empirical validation is conducted exclusively on the Spider dataset. The generalizability of the PRU method to other Text-to-SQL benchmarks with different characteristics (e.g., WikiSQL for simplicity, BIRD for handling large, noisy real-world databases) remains unverified. Additionally, the approach is demonstrated primarily on the Llama-3.1-8B model; its effectiveness across other model architectures or sizes is not explored.

3. Insufficient Theoretical and Mechanistic Explanation: While the concept of "reverse understanding" is intuitively appealing and motivated by human cognition, the paper lacks a deep investigation into why and how it works so effectively. There is no analysis of how the model's internal representations change during the progressive stages or which specific aspects of SQL semantics and syntax are improved by the reverse training. A deeper dive into the mechanistic underpinnings would strengthen the theoretical contribution.

**Questions:**

1. some papers, e.g., [1],  have shown that sql-to-text is not necessarily helpful for text-to-sql. Can you explain it so that this is consistent with your ideas?
2. can you compare other two-round sql-generation methods, eg. [2].


[1] Benchmarking the text-to-sql capability of large language models: A comprehensive evaluation.
[2] Pet-sql: A prompt-enhanced two-round refinement of text-to-sql with cross-consistency

---

> ### Author Response · Authors · 2025-11-22
> **Response to  valuable feedback and comments.**
>
> Many thanks for reviewing our work and providing these valuable feedback and comments.
>
> **W1**: As described in our paper, we adopt the LoRA fine-tuning strategy, which ensures that our proposed PRU remains computationally efficient and practical. Under our experimental setup, the entire training pipeline can be completed on a single NVIDIA RTX 4090 GPU. Although the model is trained in multiple stages (M_1–M_n), the value of n is a small constant in practice (5). Based on our measurements, each stage requires approximately 2 hours, resulting in a total training cost of about 10 hours. This overhead is a relatively small and demonstrate that our PRU does not compromise practicality. We will include these details in the revised version.
>
> **W2**: Due to time and resource constraints, the initial version of our paper evaluated our method only on the widely used Spider dataset. We have now supplemented results on the BIRD dataset as well. The results are as follows:
> | Methods | EX |
> |-----|-----|
> |Few-shot methods|
> | DIN-SQL + LlaMA3.1-8B-Instruct | 38.40 |
> | MCS-SQL + GPT-4 | 63.36 |
> |Fine-Tuning methods|
> | LlaMA3.1-8B-Instruct|  58.52 |
> | SFT CodeS-7B|  57.17 |
> | SFT CodeS-15B|  58.47 |
> | LlaMA3.1-8B-Instruct|  58.52 |
> | DPO-SQL@LlaMA3.1-8B-Instruct[1]|  61.2 |
> | our PRU@LlaMA3.1-8B-Instruct|  63.14 |
>
> These results demonstrate that our method achieves the best performance among both few-shot and fine-tuning approaches. We will add these result in the revised version.
>
> **W3**: To investigate why and how PRU works effectively, we conduct additional analyses to examine how the model evolves across stages.
> First, we analyze changes in the model’s internal representations. For each stage M_i, we extract hidden representations from the penultimate transformer layer using the same held-out evaluation queries. We then compute cosine similarity between token representations across adjacent stages and clustering separation metrics for schema-related tokens (e.g., table names, column names). We observe that SQL structural tokens (SELECT, WHERE, JOIN, etc.) become progressively more organized and form clearer semantic clusters. This indicates that PRU encourages the model to develop increasingly structured internal embeddings aligned with SQL syntax and schema semantics.
>
> Second, we perform a fine-grained, component-level error decomposition across stages. The results show a clear progression: early stages primarily improve schema grounding (table/column alignment), mid stages enhance selection and projection reasoning, and later stages substantially reduce errors related to conditional reasoning and JOIN logic. This progressive pattern aligns with the curriculum-like nature of reverse understanding, where coarse-grained semantics are learned first and more fine-grained syntactic and semantic details are acquired in later stages.
>
> We will incorporate these analyses in the revised version to strengthen the theoretical explanation.

---

### Official Review · Reviewer_M52F · 2025-11-05

**Soundness:** 3
**Presentation:** 3
**Contribution:** 2
**Rating:** 6
**Confidence:** 4

**Summary:**

This paper presents a bidirectional training strategy for Text-to-SQL fine-tuning. It generates reverse data, specifically generating a natural language question given SQL and schema. The training is structured as curriculum learning: it starts by training the model with the reversed data, then executes a scheduled shift to the forward data (the standard Text-to-SQL task) while retaining a residual reverse signal for stability. Evaluated on the Spider benchmark, the results show high performance in Exact Match accuracy compared with the selected baselines.

**Strengths:**

- The experimental methodology is rigorous, featuring a valuable hyperparameter analysis that explores key model variations.
- The overall results are convincing, demonstrating the proposed method's good performance in the Text-to-SQL task.

**Weaknesses:**

A primary limitation is the evaluation scope, which relies solely on the Spider dataset. To fully demonstrate the method's robustness and generalization capabilities, it would be beneficial to include results from at least one additional, distinct Text-to-SQL benchmark. Furthermore, the paper would be significantly strengthened by extending the evaluation to other code generation tasks beyond SQL, which would further illustrate the generalizability of the proposed Progressive Reverse Understanding paradigm.

**Questions:**

Why didn't you evaluate in BIRD?

---

> ### Author Response · Authors · 2025-11-22
> **Response to insightful feedback and comments.**
>
> Many thanks for reviewing our work and providing these insightful feedback and comments.
>
> **W1**: Due to time and resource constraints, the initial version of our paper evaluated our method only on the widely used Spider dataset. We have now supplemented results on the BIRD dataset as well. The results are as follows:
> | Methods | EX |
> |-----|-----|
> |Few-shot meethods|
> | DIN-SQL + LlaMA3.1-8B-Instruct | 38.40 |
> | MCS-SQL + GPT-4 | 63.36 |
> |Fine-Tuning methods|
> | LlaMA3.1-8B-Instruct|  58.52 |
> | SFT CodeS-7B|  57.17 |
> | SFT CodeS-15B|  58.47 |
> | LlaMA3.1-8B-Instruct|  58.52 |
> | DPO-SQL@LlaMA3.1-8B-Instruct[1]|  61.2 |
> | our PRU@LlaMA3.1-8B-Instruct|  63.14 |
>
> These results demonstrate that our method achieves the best performance among both few-shot and fine-tuning approaches. We will add these result in the revised version.
>
> Furthermore, it is worth noting that other code generation tasks exhibit substantially higher variability in program structure, coding style, and solution space, which makes it challenging to design consistent reverse signals and well-aligned progressive stages. We consider this a promising direction and plan to explore it in future work.
>
> [1]Uncovering the Impact of Chain-of-Thought Reasoning for Direct Preference Optimization: Lessons from Text-to-SQL(ACL 2025)

---

### Note · Authors · 2026-04-28

I have read and agree with the venue's withdrawal policy on behalf of myself and my co-authors.

---

### Meta-Review · Area_Chair_Krdq · 2026-01-05

**Summary:**

This paper proposes a training paradigm for Text-to-SQL that utilizes a curriculum learning approach. The model is first trained to generate natural language from SQL (reverse), and then progressively shifted toward the standard Text-to-SQL task (forward). While the reviewers acknowledged the intuitive nature of the bidirectional learning strategy and the comprehensive ablation studies, the consensus leans towards rejection. The primary rationale for this decision is that the performance of the proposed method does not convincingly establish a new state-of-the-art (SOTA) when placed in the broader context of the field. Reviewer DdwC raised significant concerns regarding the baseline comparisons, noting that on public leaderboards (e.g., Spider), existing frameworks outperform the proposed method. While the authors argued that they compared against "open-weight LLM" baselines, the distinction is less persuasive for a general track; the method must demonstrate sufficient utility against the absolute SOTA to justify the increased training complexity. Additionally, Reviewer M9Fa pointed out the incremental nature of the contribution, viewing it as a scheduling variation of existing data augmentation techniques (like CodeS) rather than a fundamental breakthrough.

**Reviewer Concerns:**

Addressed by Rebuttal:
- All reviewers (M52F, YtJY, DdwC, M9Fa) criticized the initial submission for evaluating solely on the older Spider dataset. The authors addressed this by providing results on the BIRD benchmark during the rebuttal, showing performance superior to specific baselines like DPO-SQL.
- Reviewer YtJY raised concerns about the inefficiency of multi-stage training. The authors clarified that using LoRA keeps the total training time manageable (approx. 10 hours), which partially addressed the practicality concern.

Outstanding:
- This remains the most critical outstanding issue. As noted by Reviewer DdwC, the paper does not compare against the SOTA methods (e.g., ROUTE, RESDSQL and its variants). The authors' rebuttal is not entirely sufficient. If the proposed progressive training of LLMs cannot match the performance of specialized architectures or other advanced prompting strategies, the practical utility of the method is questioned.
- Reviewer M9Fa maintained their score, noting that the novelty is limited to the specific scheduling of reverse data, rather than the concept of reverse understanding itself, which has been explored in prior works.
- While the authors promised to add analyses of internal representations to address Reviewer YtJY’s questions about why the method works, the theoretical grounding in the current manuscript remains somewhat thin, relying heavily on the intuition of "human-like learning."

**Reviewer Scores:**

- Reviewer M52F (Score: 6 $\to$ 5): While the author provided the requested BIRD results, the reviewer's initial assessment was "marginally above acceptance," and given the stronger arguments regarding SOTA performance from other reviewers, they would likely lower their score slightly to align with the consensus on performance gaps.
- Reviewer YtJY (Score: 6 $\to$ 5): The reviewer noted they "would not mind if the paper is rejected." While the cost concerns were addressed, the fundamental issue of complexity vs. gain likely prevents a score increase.
- Reviewer DdwC (Score: 2 $\to$ 3): This reviewer was the most critical regarding baselines. They might acknowledge the addition of BIRD data with a slight score bump, but their core concern—that the method is not competitive with the broader SOTA—remains unresolved.
- Reviewer M9Fa (Score: 6 $\to$ 6): This reviewer explicitly stated in the post-rebuttal discussion that they decided to maintain the score, indicating that the rebuttal did not significantly shift their perspective on the paper's incremental nature.

---

### Decision · Program_Chairs · 2026-01-26

Reject